# Elucidation of the role of metals in the adsorption and photodegradation of herbicides by metal-organic frameworks

Nan Chieh Chiu[1,4], Jacob M. Lessard[1,4], Emmanuel Nyela Musa[1,4], Logan S. Lancaster [2], Clara Wheeler[2], Taylor D. Krueger[2], Cheng Chen[2], Trenton C. Gallagher[2], Makenzie T. Nord [2], Hongliang Huang [3] ✉, Paul Ha-Yeon Cheong [2] ✉, Chong Fang [2] ✉ & Kyriakos C. Stylianou [1] ✉

Here, four MOFs, namely Sc-TBAPy, Al-TBAPy, Y-TBAPy, and Fe-TBAPy (TBAPy: 1,3,6,8-tetrakis(*p*-benzoic acid)pyrene), were characterized and evaluated for their ability to remediate glyphosate (GP) from water. Among these materials, Sc-TBAPy demonstrates superior performance in both the adsorption and degradation of GP. Upon light irradiation for 5 min, Sc-TBAPy completely degrades 100% of GP in a 1.5 mM aqueous solution. Femtosecond transient absorption spectroscopy reveals that Sc-TBAPy exhibits enhanced charge transfer character compared to the other MOFs, as well as suppressed formation of emissive excimers that could impede photocatalysis. This finding was further supported by hydrogen evolution half-reaction (HER) experiments, which demonstrated Sc-TBAPy's superior catalytic activity for water splitting. In addition to its faster adsorption and more efficient photodegradation of GP, Sc-TBAPy also followed a selective pathway towards the oxidation of GP, avoiding the formation of toxic aminomethylphosphonic acid observed with the other $M^{3+}$-TBAPy MOFs. To investigate the selectivity observed with Sc-TBAPy, electron spin resonance, depleted oxygen conditions, and solvent exchange with $D_2O$ were employed to elucidate the role of different reactive oxygen species on GP photodegradation. The findings indicate that singlet oxygen ($^1O_2$) plays a critical role in the selective photodegradation pathway achieved by Sc-TBAPy.

Glyphosate (GP), the active ingredient in commercial weed-killers such as Round-Up®, is the world's most widely used herbicide[1–3]. Its persistence in soil ranges from days to months, increasing if it reaches water sources[4–8], and GP has been detected in groundwater, surface water, and human urine[8–11]. Exposure to GP is associated with health risks, including an increased risk of non-Hodgkin's lymphoma and multiple myeloma[12–15], as well as developmental defects in animals[16,17], prompting the Environmental Protection Agency (EPA) to set a Maximum Contaminant Level Goal (MCLG) of 700 ppb[17–20].

Environmental degradation of GP occurs primarily via two bacterial pathways: GP oxidoreductase (GOX) cleaves the C–N bond, forming glyoxylate and aminomethylphosphonic acid (AMPA)[21–23],

[1]Materials Discovery Laboratory (MaD Lab), Department of Chemistry, Oregon State University, 153 Gilbert Hall, Corvallis, OR 97331, USA. [2]Department of Chemistry, Oregon State University, 153 Gilbert Hall, Corvallis, OR 97331, USA. [3]State Key Laboratory of Separation Membranes and Membrane Processes, School of Chemical Engineering and Technology, Tiangong University, 300387 Tianjin, China. [4]These authors contributed equally: Nan Chieh Chiu, Jacob M. Lessard, Emmanuel Nyela Musa. ✉e-mail: huanghongliang@tiangong.edu.cn; cheongh@oregonstate.edu; chong.fang@oregonstate.edu; kyriakos.stylianou@oregonstate.edu

while C–P lyase cleaves the C–P bond, yielding benign phosphate and sarcosine[23–26]. AMPA exhibits similar toxicity and a longer half-life than GP, interferes with DNA synthesis and repair in fish and amphibians, and can have adverse effects on human blood cells[24–28], making the C–P lyase pathway more desirable[21,29–31]. Bacterial and fungal species have been studied for biodegradation of GP in water[21,32], however, the performance of microorganisms for GP remotion depends on factors such as pH, temperature, and GP concentration, as well as on deficits of available nitrogen and phosphorus, which are not typical in natural environments[30].

Adsorptive materials can remove GP from water via hydrogen-bond interactions, electrostatic interactions, and complexation[30,33]. Materials studied for GP adsorption include activated carbon, bio-carbon, graphene oxide, alum sludge, goethite, and zeolite[33–37]. The primary disadvantage of these materials is that recovery of adsorbates is challenging, and accumulation of adsorbates over time reduces their efficiency[30]. Moreover, the high efficiency of GP removal has often required acidic conditions, as the net charge of porous materials and GP become more negative with increased pH, thereby increasing electrostatic repulsion and hindering adsorption[30,38].

Oxidative processes for GP remotion include photo-Fenton and electro-Fenton oxidation, photolysis using hydrogen peroxide or ozone in combination with UV light, electrochemical oxidation, and photocatalysis[24,39–48]. However, the majority of these strategies result in the formation of toxic AMPA. Among these approaches, photo-catalysis is promising because it operates under mild conditions, is considered a green technology, and can exhibit high selectivity[49–51]. Holes ($h^+_{VB}$) generated in the valence band (VB) of the catalyst can directly oxidize organic pollutants[52], or oxidize water to form hydroxyl radicals (•OH), while excited electrons ($e^-_{CB}$) in the conduction band (CB) can reduce dissolved oxygen ($O_2$) to generate superoxide anion radicals (•$O_2^-$), which can undergo oxidation by $h^+_{VB}$ to form singlet oxygen ($^1O_2$)[53]. This array of reactive oxygen species (ROS) affords a diverse set of reaction pathways for degrading molecules, providing the potential for GP degradation which avoids the formation of AMPA.

The majority of studies on photocatalytic degradation of GP employ the metal oxide semiconductor $TiO_2$, with a few studies focused on $MnO_2$[30]. Recently, our group demonstrated complete GP degradation in 5 mM (845 ppm) GP solution after 8 h of irradiation using a nitrogen and sulfur-doped $TiO_2$ anatase (NSTA) derived from the metal-organic framework (MOF) MIL-125-$NH_2$[54]. While kinetic investigations revealed that AMPA was formed initially, AMPA was almost entirely degraded after the reaction. We concluded that the superior activity of NSTA compared to other phases in our study was due to its higher porosity, smaller band gap, and the presence of oxygen vacancies etc.

To address the current limitations of GP remediation, we envisioned a material that combines rapid adsorption with efficient and selective photocatalytic degradation capacity. MOFs are a versatile class of porous materials composed of metal clusters linked by organic ligands, and can be tailored for small molecule capture and photo-degradation by strategically selecting metals and ligands to tune their structural, chemical, and optical properties[55–66]. Along with their broad tunability, their large surface areas and abundant active sites make MOFs theoretically superior to traditional adsorbents and photo-catalysts for GP removal. To date, research involving MOFs for GP remediation has focused primarily on adsorption[58,67–69], with few studies involving MOFs for photodegradation of GP[70]. NU-1000, UiO-66, UiO-67, MIL-125, and MIL-101 have been shown as effective MOFs for GP adsorption, with NU-1000 exhibiting exceptional adsorption perfomance[58,67–69]. It is noteworthy that the pyrene-based 1,3,6,8-tet-rakis(*p*-benzoic acid)pyrene (TBAPy) ligands of NU-1000 are complementary to photocatalysis, imparting efficient energy transfer, and a long excited-state lifetime[71–73].

Inspired by the structure of NU-1000, we selected four iso-structural $M^{3+}$-TBAPy MOFs to investigate the adsorption and photo-degradation of GP. We reasoned that Lewis acidity of the $M^{3+}$ ions would play a direct role in promoting affinity for GP capture via interaction with its Lewis basic phosphonic acid group, and that GP photodegradation would be promoted by the long excited-state life-time and electron hole-pair dissociation provided by the TBAPy ligands. Furthermore, the narrow channels and sandwiched pyrene cores of the $M^{3+}$-TBAPy MOFs can promote host-guest interactions, and these types of MOFs have proven capable of efficient and selective photooxidation of organic molecules[74]. Utilizing an isostructural family of MOFs based on trivalent metals allows for investigating the influence of MOF metal variation on adsorption on photodegradation processes. Varying the metal nodes is a powerful tool in MOF design, as metal properties such as size, electronegativity, and softness/hardness influence Lewis acidity[75], which in turn affects metal-ligand synergy, as interactions between *d* orbitals of different transition metals and ligand orbitals influences ligand-to-metal charge transfer (LMCT)[76]. Metal variation, therefore, enables tuning of band structure, charge separation, excited-state lifetime, and light-harvesting properties. Enhanced charge separation and excited-state lifetimes allow for more efficient utilization of $e^-_{CB}$ and $h^+_{VB}$, while tuning band edge potentials can be used for controlling ROS generation[53], and for aligning the MOF VB potential with the highest occupied molecular orbital (HOMO) of GP to promote oxidation. We hypothesized that changing the metal ions in $M^{3+}$-TBAPy MOFs would provide information regarding the effects of metal-ligand synergy on GP adsorption and serve as a strategy to investigate selective photocatalytic mechanisms for GP degradation.

## Results

### Synthesis and characterization

We synthesized and characterized four isostructural MOFs, namely $M^{3+}$-TBAPy with formula of [$M_2(OH)_2$(TBAPy)], where $M^{3+}$ represents $Al^{3+}$, $Fe^{3+}$, $Sc^{3+}$, and $Y^{3+}$[77]. These MOFs feature a structure composed of chains of octahedral M(III)-$O_4(OH)_2$ units, with each M(III) coordinated to four TBAPy ligands and two $\mu_2$ trans hydroxide anions (Fig. 1a)[77]. The chains of M(III)-$O_4(OH)_2$ units align parallel to the *b* axis within the MOF structure (Fig. 1b). The arrangement of hydroxide and TBAPy carboxylate groups alternate on opposite sides of the M(III)-$O_4(OH)_2$ chain, which is related by the $2_1$-screw axis of the $C_{mmm}$ space group (Fig. 1c). The orientation of TBAPy ligand around the one-dimensional chains leads to the generation of three distinct pores (Fig. 1). Powder X-ray diffraction (PXRD) analysis confirmed the formation and purity of the $M^{3+}$-TBAPy MOFs, indicating that they are isostructural (Fig. 2a). Thermogravimetric analysis (TGA) revealed the thermal stability of all $M^{3+}$-TBAPy analogues up to 300 °C (Supplementary Fig. 1). Nitrogen ($N_2$) isotherms conducted at 77 K and 1 bar demonstrated the micro-porous nature of all MOFs, with Brunauer−Emmett−Teller (BET) surface areas of 793, 975, 1119, and 1055 $m^2$ $g^{-1}$ for Sc-TBAPy, Y-TBAPy, Al-TBAPy, and Fe-TBAPy, respectively (Fig. 2b). Electrochemical impedance spectroscopy (EIS) was employed to evaluate the electrical conductivity of each MOF. As shown in Fig. 2c, Fe-TBAPy, Al-TBAPy, and Y-TBAPy exhibit comparable charge transfer resistance. Sc-TBAPy shows the largest semicircle radius in the EIS plot which represents the highest charge transfer resistance property among all four MOFs.

### Photophysical properties

The photophysical properties of each MOF were investigated using ultraviolet−visible (UV−vis) spectroscopy and X-ray photoelectron spectroscopy (XPS). Results from the UV−vis spectroscopic studies revealed that Fe-TBAPy exhibits an absorption edge at ~600 nm. Y-TBAPy and Sc-TBAPy display a similar absorption edge close to 470 nm, while Al-TBAPy showed an absorption edge around 400 nm (Fig. 2d). Energy bands, including the VB, CB and bandgap ($E_g$), are

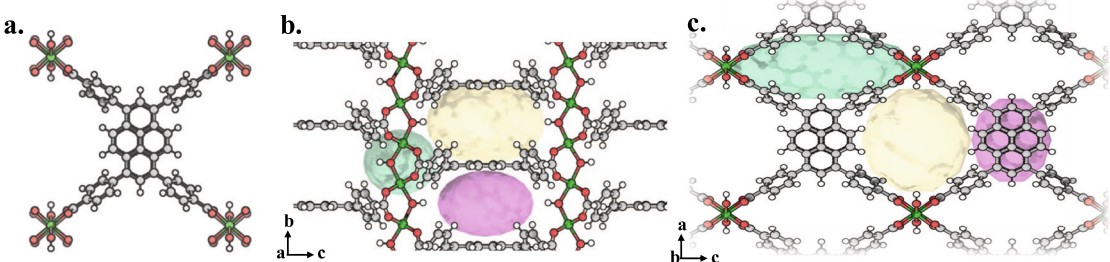

**Fig. 1 | Structure representation of the M³⁺-TBAPy, [M₂(OH)₂(TBAPy)], family of MOFs, including Sc-TBAPy, Al-TBAPy, Y-TBAPy, and Fe-TBAPy. a** View of the TBAPy ligand coordinated to eight neighboring M³⁺ atoms. **b, c** Packing of the structure viewed along a and b directions, respectively. The orientation of the TBAPy ligand around the M³⁺-chains leads to the generation of M³⁺-TBAPy with three-dimensional pores: pore A (pink), pore B (yellow), and pore C (green). Atom color code: green, M³⁺; gray, C; red, O; white, H.

crucial properties of photocatalytic materials. The bandgap was calculated from the diffuse reflectance spectroscopy data. Al-TBAPy has the largest bandgap (2.78 eV), followed by Sc-TBAPy (2.58 eV) and Y-TBAPy (2.52 eV), while Fe-TBAPy exhibits the smallest bandgap (1.86 eV) among the four MOFs (Fig. 2d and Supplementary Table 2). Because the M³⁺ metals of the MOFs are electronically deficient, photogeneration of charge carriers ($e^-_{CB}$ and $h^+_{VB}$) is expected to proceed through LMCT[78]. As the energy required to initiate LMCT decreases with red-shifted absorption edges and smaller bandgaps[79], UV-vis experiments suggest that the LMCT is most favorable in the order of Fe-TBAPy > Y-TBAPy > Sc-TBAPy > Al-TBAPy. XPS analysis with linear extrapolation method provided insights into the experimental VB for all MOFs[80,81], and the CB of each MOF was determined by subtracting the bandgap energy from the VB potential energy (Supplementary Figs. 2–4, and Supplementary Table 2). The CB values for all MOFs (Sc-TBAPy: −0.19 eV, Y-TBAPy: −0.02 eV, Al-TBAPy: −0.06 eV, and Fe-TBAPy: −0.82 eV) are more negative than the reduction potential of water (H⁺/Hydrogen (H₂), 0.0 eV vs. SHE (standard hydrogen electrode)). This result indicates that the CB of all four MOFs are thermodynamically favored for water reduction and H₂ production, making them suitable for water splitting applications. On the other hand, the VB values of Sc-TBAPy, Y-TBAPy, and Al-TBAPy are more positive than the oxidation potential of water (O²⁻/O₂, 1.23 eV vs. SHE). Only the VB of Fe-TBAPy is less positive than the oxidation potential of water. The difference in VB levels among the MOFs can influence the thermodynamic driving force for hole transfer ($h^+_{VB}$) and the generation of ROS, thus affecting the oxidation half-reaction. PDOS computations for VB, CB, and $E_g$ follow the same trend as the experimentally determined values, supporting the photophysical properties elucidated from UV-vis and XPS (Supplementary Table 2, Supplementary Figs. 2–4). Additionally, CB energy levels for Al-TBAPy and Sc-TBAPy were further verified through Mott−Schottky plots with measurements performed at frequencies of 500, 1000, and 1500 Hz (Supplementary Fig. 5).

To understand the influence of the excited-state electronic dynamics on redox reactions, fs-TA experiments were performed on our four MOFs suspended in solution (Fig. 3) after steady-state electronic spectroscopy (Supplementary Fig. 6). The fs-TA spectra, aided by global analysis, were used to delineate the evolution and decay-associated difference spectra (EADS and DADS) of four MOFs in a systematic manner upon 400 nm excitation (Supplementary Figs. 7, 8). After a few picoseconds, the fs-TA spectra of Sc-TBAPy exhibit a distinct peak around 580 nm (indicated by an asterisk in Fig. 3a) with a longer lifetime compared to the other peaks. This newly formed peak can be associated with a charge transfer (CT) state, which has been proposed for the H₄TBAPy ligand with our detailed spectral characterization (Supplementary Fig. 9)[82-86]. However, this CT band was not clearly observed in the other three MOFs due to overlap with a negative feature below 600 nm. This negative feature can be attributed to stimulated emission (SE) as it falls within the steady-state emission region (Supplementary Figs. 6b, 9a), and there is no significant ground-state absorbance above 450 nm (Supplementary Figs. 6a, 9a). Notably, the CT band in Al-TBAPy grows more slowly, reaching its maximum at ~100 ps (Fig. 3b), while Fe-TBAPy exhibits only a small positive signal without a clear CT band (Fig. 3c). Y-TBAPy takes less time (~25 ps) compared to Al-TBAPy to reach its CT band maximum (Fig. 3d). Although the different decay rates of the initial SE bands may affect the apparent emergence of CT band in a similar spectral region, the overall prominence of the CT band in Sc-TBAPy and Al-TBAPy is notable when compared with Fe-TBAPy and Y-TBAPy.

At longer time delays (>1 ns), a broad SE band becomes dominant in the spectrum of Fe-TBAPy (around 550 nm) and Y-TBAPy (around 500 nm) (Fig. 3c, d), while it is reduced in Al-TBAPy (Fig. 3b) and barely observable in Sc-TBAPy (Fig. 3a). This late SE band, supported by the TA spectra of the ligand (Supplementary Fig. 9), is attributed to the formation of emissive excimers. Excimers could form in MOFs with densely packed linkers in highly ordered arrays. These emissive species may occur within the MOFs via interchromophoric interactions[87-89]. Notably, the rise of the excimer SE band in the MOFs is considerably slower compared to the aggregated state of the ligand (Supplementary Fig. 9d, ~300 ps retrieved for the suspended ligand in acetonitrile (ACN)) and exhibits significant variation among the MOFs (Supplementary Figs. 6, 7, 8). The formation of excimer occurs in the following order from fastest to slowest: Y-TBAPy (460 ps) > Fe-TBAPy (650 ps) > Al-TBAPy (750 ps) > Sc-TBAPy (around 1.5 ns). The extended time for excimer formation in MOFs can be explained by the restricted movements of the coordinated TBAPy linkers, which undergo conformational changes to facilitate emission, similar to the ligand in ACN suspension (Supplementary Fig. 9d). Meanwhile, the shorter emission lifetime of the excimer indicates an increased nonradiative decay of H₄TBAPy ligands. This result is attributed to greater pyrene-pyrene distances in the MOF as well as increased solvent interactions, thus promoting the quenching of the excimer state[87]. Detailed analysis and additional discussion regarding fs-TA spectral data can be found in the Supporting information. This important finding explains the minimal broadening of fluorescence observed in the MOFs since nonradiative pathways dominate over fluorescence as the primary relaxation pathway for MOF excimers.

Overall, our results from the fs-TA spectra indicate that Sc-TBAPy exhibits enhanced CT character compared to the other MOFs, as well as suppressed formation of emissive excimers that could passivate photogenerated carriers and hinder photochemistry. The differences in metal nodes (Fe, Y, Al, and Sc) within these pyrene-based MOFs demonstrate how the interaction strength between the TBAPy linkers is influenced by the specific metal, thus impacting the potential energy landscape and relaxation pathways after electronic excitation. Through a comprehensive analysis using the free H₄TBAPy ligand and metal-dependent MOFs in suspension, we have elucidated the

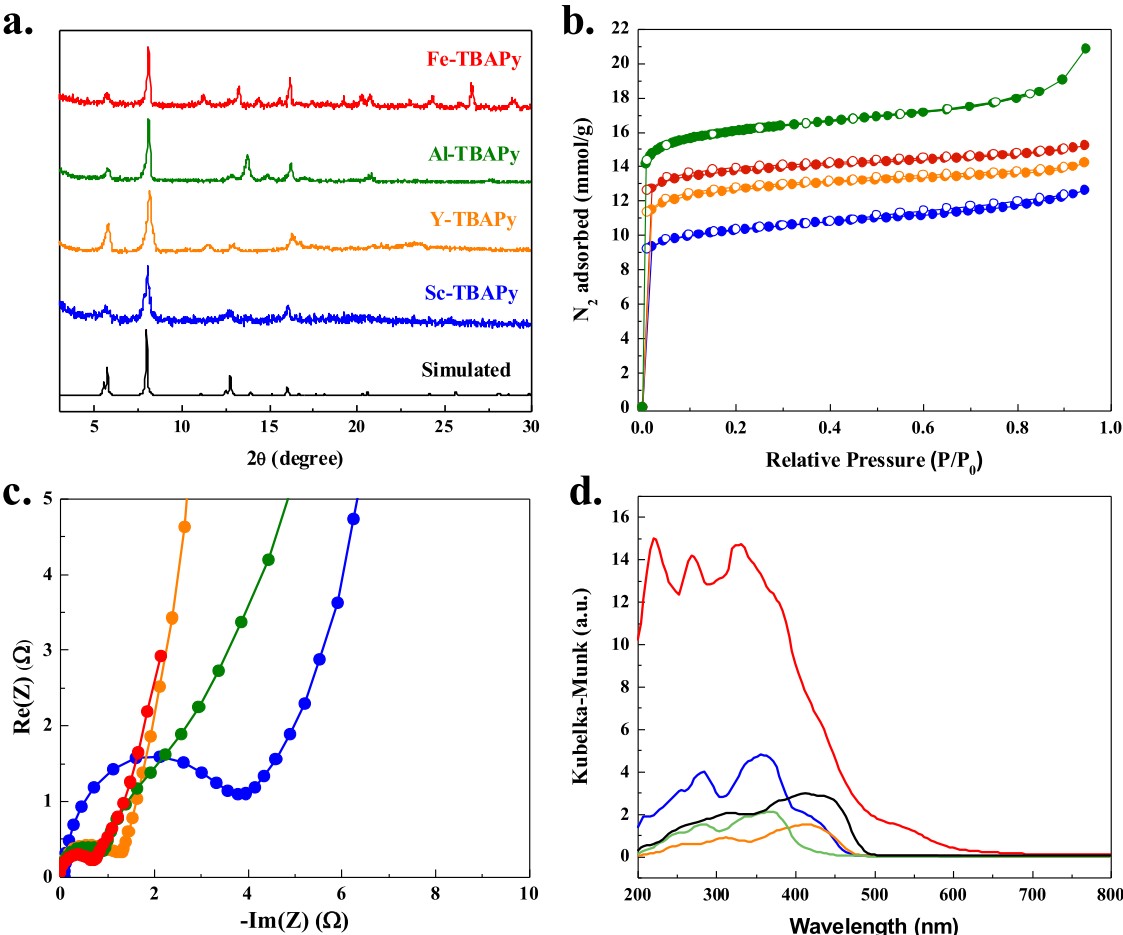

**Fig. 2 | Solid-state characterization of the four M³⁺-TBAPy MOFs. a** Powder X-Ray diffraction (PXRD) patterns. All PXRD patterns of MOFs match the simulated M³⁺-TBAPy PXRD pattern (black), indicating that they are isostructural and can be synthesized as phase pure. **b** Type I $N_2$ isotherms collected at 77 K and 1 bar reveal that all MOFs are microporous. Filled symbols represent adsorption and empty symbols represent desorption. **c** EIS Nyquist plots reveal that Sc-TBAPy (blue) has the largest semicircle radius, representing the highest charge transfer resistance. **d** Kubelka–Munk transformation diffuse reflectance spectra collected from 200 to 800 nm. Black line corresponds to the absorption spectrum of the protonated ligand, $H_4$TBAPy.

emergence of an intrinsic CT process that likely plays a functional role for the photocatalytic activity of M³⁺-TBAPy at the macroscopic level.

## Hydrogen evolution half-reaction

After investigating the influence of different metal nodes on the photophysical properties of the MOFs, hydrogen evolution half-reaction (HER) experiments were conducted. It was anticipated that the more photocatalytically active materials would generate more $e^-_{CB}$ upon light irradiation, leading to enhanced utilization of the $e^-_{CB}$ for optimal $H_2$ production. Among the tested MOFs (mixed with $Ni_2P$ as a co-catalyst, and triethylamine as a sacrificial reagent), Sc-TBAPy exhibits the highest $H_2$ evolution rate of approximately 556 μmol h⁻¹ g⁻¹, while Y-TBAPy, Al-TBAPy and Fe-TBAPy display inferior catalytic activity with HER rates of 42, 8 and 0 μmol h⁻¹ g⁻¹, respectively (Fig. 4a). Except for Fe-TBAPy, all MOFs maintain their crystallinity after photocatalysis. The superior photocatalytic activity of Sc-TBAPy compared to the other M³⁺-TBAPy is supported by the aforementioned fs-TA studies, revealing a clear correlation between the variation of metal nodes (Fe, Y, Al, and Sc) in the pyrene-based MOFs and the resultant modulation of linker-linker interaction strength. Interestingly, despite Fe-TBAPy having the smallest bandgap among all the MOFs, it exhibits the lowest HER activity. The underlying reason is its low structural stability confirmed by the post-HER PXRD. In contrast, Sc-TBAPy, Y-TBAPy and Al-TBAPy retain their crystalline structures after HER (Supplementary Fig. 10).

## Photocatalytic degradation of GP

After studying the HER activity, we proceeded to investigate the oxidizing effects of the M³⁺-TBAPy MOFs for the photocatalytic degradation of GP. As mentioned above, the optical properties of the M³⁺-TBAPy indicate their abilities to oxidize GP due to their VB positions: Sc at 2.39, Al at 2.72, and Y at 2.50 eV (Supplementary Fig. 2 and Supplementary Table 2), all of which are more positive than the HOMO of GP (1.05 eV vs SHE)[90]. Only Fe-TBAPy, with VB at 1.04 eV, falls slightly less positive than the calculated HOMO of GP, implying that Fe-TBAPy may not be suitable for GP oxidation. Notably, based on the superior HER activity observed for Sc-TBAPy, attributed to its fast CT kinetics and longer charge lifetime, we hypothesized that Sc-TBAPy would exhibit the best performance for GP oxidation. In other words, a better separation between $h^+_{VB}$ and $e^-_{CB}$ would result in the generation of a significant amount of active ROS driving the oxidation reaction. To evaluate their oxidative potentials, we prepared a 1.5 mM aqueous solution of GP and subjected it to irradiation with the M³⁺-TBAPy MOFs. This concentration was determined suitable to analyze the concentration of GP as it decreases in solution. The products of GP oxidation were further identified using ¹H, ¹³C and ³¹P NMR analysis.

Analysis of the ¹H NMR spectra revealed that GP oxidation commences within 5 min of irradiation, with all the M³⁺-TBAPy MOFs demonstrating varying activities (Fig. 4b). However, upon continuous irradiation for 8 h, an increase oxidative activity was observed with only 11.81% GP remaining in solution using Fe-TBAPy as a

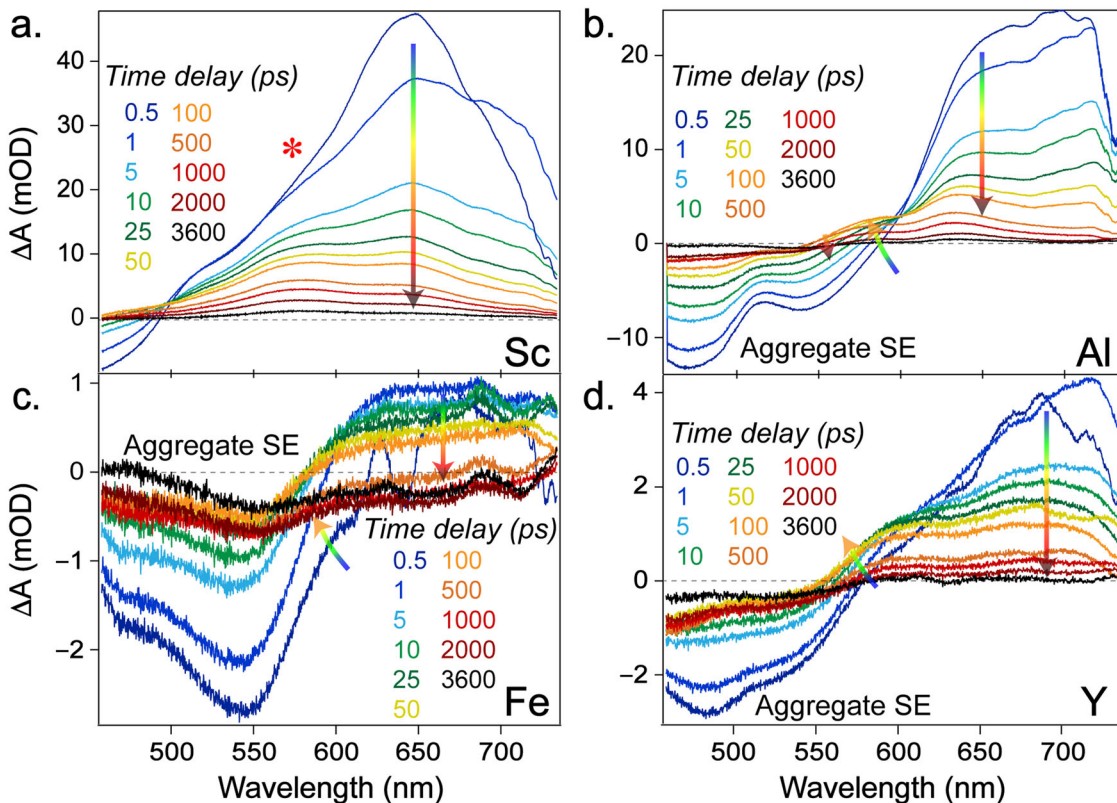

**Fig. 3 | Transient electronic spectra of MOFs after 400 nm excitation.** Selective time points from the fs-TA spectra for (**a**) Sc-TBAPy, (**b**) Al-TBAPy, (**c**) Fe-TBAPy, and (**d**) Y-TBAPy, suspended in DMF, are plotted with the spectral evolution denoted by the color-gradient arrows representing early-to-late time points. The prominent stimulated emission (SE) bands from the aggregated ligand state are denoted in (**b**–**d**).

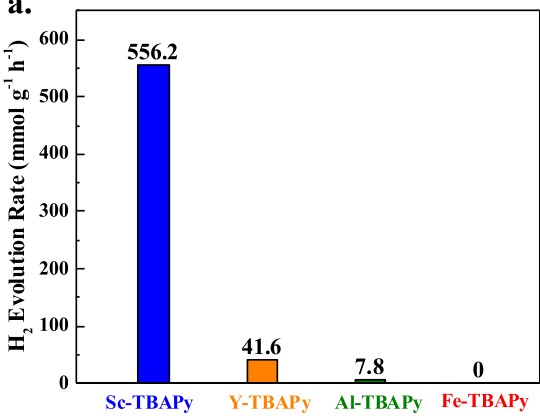

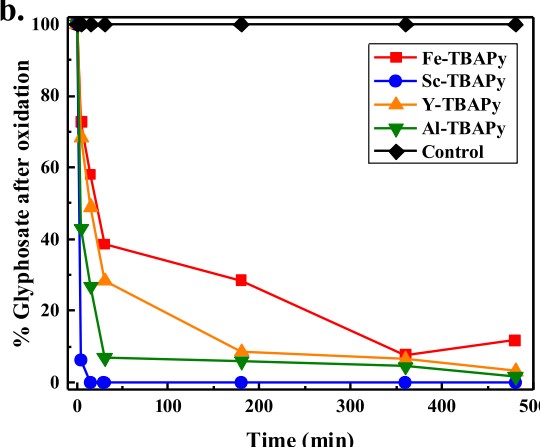

**Fig. 4 | Photocatalytic activity of MOFs. a** Hydrogen evolution rates measured using Sc-TBAPy (blue), Y-TBAPy (yellow), Al-TBAPy (green), and Fe-TBAPy (red) photocatalyst. Similar activities were observed with Y-TBAPy and Al-TBAPy, showing 3.16% and 1.67% of GP remaining in solution, respectively after 8 h (Fig. 4b). Importantly, when the reaction was performed using Sc-TBAPy, no evidence of GP was observed after 8 h of irradiation. To assess the stability of the MOFs after photocatalysis, PXRD analysis confirmed that all four MOFs remain stable and recyclable (Supplementary Fig. 11).

Further analysis of the $^1$H NMR spectra (Supplementary Fig. 12) over the different time courses indicated progressive oxidation of GP, leading to the formation of mixed products such as AMPA, formic acid, glycine, and acetic acid. After 5 min irradiation with Fe-TBAPy, 10.80% with triethylamine (TEA) as a sacrificial reagent and Ni$_2$P as co-catalysts. **b** Oxidation of GP studied under oxic conditions in H$_2$O.

formic acid and 2.14% acetic acid were observed in solution along with a high concentration of GP (Fig. 5c). However, after 3 h, AMPA and glycine emerged, resulting in a drop in GP concentration; 88.19% total GP degradation was achieved after 8 h (Fig. 5c). The oxidation of GP with Y-TBAPy and Al-TBAPy followed a similar trend, generating various products as observed with Fe-TBAPy, and achieving total GP conversions of 96.84% and 98.33% after 8 h, respectively (Fig. 5b–d). In the case of Sc-TBAPy, analysis of the $^1$H NMR spectra (Supplementary Fig. 12a) collected after 5 min revealed the formation of only formic acid (1.21%) in the solution. Upon extended irradiation, we observed the emergence of glycine, with no trace of GP and AMPA after 3 h.

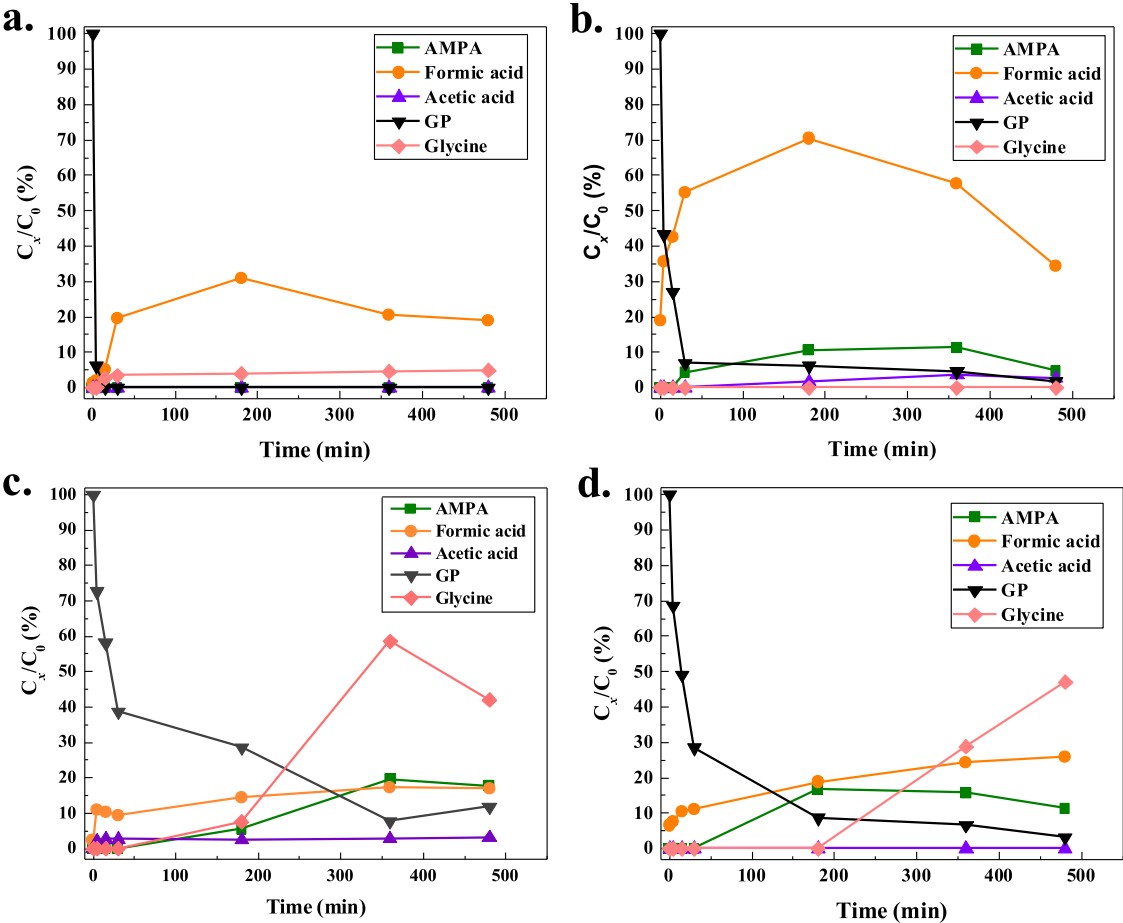

**Fig. 5 | Key product formation upon GP degradation by. a** Sc-TBAPy, (**b**) Al-TBAPy, (**c**) Fe-TBAPy, and (**d**) Y-TBAPy, under oxic conditions in $H_2O$. All MOFs degrade GP, with Sc-TBAPy exhibiting superior and selective catalytic activity in the degradation of GP to glycine and formic acid.

Interestingly, further irradiation of the GP solution up to 8 h resulted in an increased conversion of GP to glycine (4.93%) and formic acid (19.22%), while GP remained undetectable in the solution (Fig. 5a). To validate the observations made through $^1H$ NMR analysis, we also employed $^{13}C$ NMR (Supplementary Fig. 13) and $^{31}P$ NMR (Supplementary Fig. 14) techniques. Based on the varying oxidative activities of all $M^{3+}$-TBAPy MOFs, Sc-TBAPy exhibits the highest photocatalytic activity for GP oxidation, matching our predictions. Notably, low yields of formic acid and glycine from GP oxidation using Sc-TBAPy suggest the presence of another mechanism driving the reaction. This led us to conduct a more detailed investigation into several factors that could potentially contribute to GP oxidation with Sc-TBAPy: (i) the adsorption of GP within the pores of Sc-TBAPy, (ii) the enhanced interactions resulting from GP's confinement in the MOF pores, and (iii) the identification of ROS under different conditions, including oxic vs. anoxic environments and $H_2O$ vs. $D_2O$ solvents.

## GP Adsorption within the Sc-TBAPy Pores

To investigate the adsorption of GP by our $M^{3+}$-TBAPy MOFs, each MOF was exposed to a 1.5 mM solution of GP for 5 min in the absence of light. Subsequent $^1H$ NMR analysis revealed that Fe-TBAPy, Y-TBAPy, and Al-TBAPy removed 2%, 17%, and 36% of GP from solution, respectively, while Sc-TBAPy completely removed GP after 5 min (Supplementary Figs. 15, 16). These experiments demonstrate that altering the metal within isostructural MOFs can dramatically affect their adsorption performance toward GP. Motivated by the rapid GP removal observed with Sc-TBAPy, we conducted molecular mechanics

simulations to investigate the potential adsorption sites and complexation energies for GP within each MOF. The results suggest that three pores (active sites), labeled as A, B and C, are plausible for GP adsorption in the $M^{3+}$-TBAPy MOFs as depicted in Fig. 1b, c. Pore A represents the intercalation site between the two planes of the MOF pyrene cores, while pore B corresponds to the region between the long edges of the pyrene, and pore C is located between the short edges of the pyrene core (Fig. 1b). Our simulations suggest that Sc-TBAPy and Al-TBAPy favor GP complexation in pore B (Supplementary Fig. 17, Supplementary Table 3). In agreement with our uptake experiments, Sc-TBAPy exhibits the highest binding energy for GP (Supplementary Table 3). These results highlight how the properties of the specific metal within our $M^{3+}$-TBAPy MOFs influences the adsorption of GP based on favorability of complexation within the MOF pores.

The rapid removal of GP by Sc-TBAPy without light prompted us to delve deeper into whether fast GP removal by Sc-TBAPy with light was due to adsorption or photodegradation. To investigate this, we first performed the reaction in the dark for three consecutive cycles. Analysis of the $^1H$ NMR data obtained after the first cycle revealed no trace of GP in the solution (Supplementary Fig. 18a). However, after the second cycle, GP remained in the solution. Additionally, upon completion of the third cycle, the $^1H$ NMR results indicated a higher concentration of GP in the solution (Supplementary Fig. 18a). This unexpected finding led us to propose that the active sites of Sc-TBAPy may have become saturated, inhibiting further GP uptake. Alternatively, when the reaction was conducted under light, no GP was detected in solution. The reaction under light was subsequently

performed over three consecutive cycles, producing glycine and formic acid with no sign of GP in solution, as confirmed by $^1$H NMR analysis (Supplementary Fig. 18b). The absence of GP in solution indicates its complete degradation and the ability of the Sc-TBAPy to photodegrade GP over three cycles. During the same oxidation experiments with Al-TBAPy in the dark, we observed a substantial increase in the GP concentration in the solution after the first cycle (Supplementary Fig. 18c). However, no further uptake of GP was observed in the second and third cycles. Upon irradiation, the formation of AMPA and formic acid was detected with GP present, consistent with our previous observations (Fig. 5). Therefore, we propose that GP is initially captured and retained within the pores of the Sc-TBAPy, ultimately reaching a saturation point in the absence of light. However, under light irradiation, the continuous generation of ROS facilitates the attack on GP, causing the cleavage of relevant bonds and the release of degradation products from the MOF's pores. Consequently, the active sites and pores of Sc-TBAPy are regenerated, enabling further adsorption and catalysis, as confirmed from the recycling experiments (Supplementary Fig. 19).

To verify that GP was not merely adsorbed within the pores of Sc-TBAPy, the MOF was recovered after the photooxidation reaction and washed with dimethylformamide (DMF) to facilitate the removal of any GP within its pores. The MOF's BET surface area was then measured, and the DMF-washed solution was analyzed. Subsequent $^1$H NMR analysis of the DMF-washed solution revealed the absence of GP (Supplementary Fig. 18c), and the measured BET surface area (Supplementary Fig. 20) was comparable to that of the pristine Sc-TBAPy. Specifically, Sc-TBAPy exhibits a BET surface area of 654 m$^2$ g$^{-1}$ after exposure to GP under UV light, 714 m$^2$ g$^{-1}$ after exposure to GP in the dark, and 832 m$^2$ g$^{-1}$ and 884 m$^2$ g$^{-1}$ after DMF washing following exposure to GP in the dark and light, respectively (Supplementary Table 4). The $^{13}$C (Supplementary Fig. 21a) and $^{31}$P (Supplementary Fig. 21b) NMR analyses supported the findings from the $^1$H NMR, confirming the absence of GP in the wash solution. Additionally, FTIR analysis was conducted to assess the stability of the Sc-TBAPy MOF after exposure to light. The FTIR spectra demonstrate that Sc-TBAPy retains its structure under both conditions (Supplementary Fig. 22). Our findings suggest that Sc-TBAPy possesses high affinity for GP, leading to a stronger interaction between GP and pore surface, which in turn enhances the adsorption of GP on Sc-TBAPy[72]. To investigate the impact of Sc-TBAPy in the degradation of GP in a natural environment, a 1.5 mM GP solution was prepared using sampled river water. Additionally, another GP solution with the same concentration was prepared, and its pH was adjusted to 10.7 using 1 M NaOH to evaluate the activity of the MOF in an oxidizing environment. Subsequently, the reactions were performed. Results from $^1$H NMR analysis revealed complete degradation of GP in the solutions, leading to the formation of glycine and formic acid (Supplementary Fig. 23). To verify the stability Sc-TBAPy under basic conditions, its PXRD pattern was collected. The results demonstrated that the recovered Sc-TBAPy remained stable after exposure to irradiation in a pH 10.7 GP solution, indicating the superior activity of the MOF in both acidic and basic environments (Supplementary Fig. 24). In addition, adsorption experiments for AMPA and glycine, two main metabolites of GP, were conducted under dark conditions using Sc-TBAPy. Analysis of the $^1$H NMR spectra revealed that a significant amount of glycine, approximately 99.4%, remained in the solution after the reaction (Supplementary Fig. 25a). However, no trace of AMPA was detected in the solution post-reaction (Supplementary Fig. 25b). Our rational for this different sorption behavior between AMPA and glycine is attributed to the phosphonic acid group present in AMPA, akin to that in GP, which enables binding to the MOF. In contrast, such binding is minimal in glycine due to the absence of the phosphonic acid group. These results further support the selective nature of Sc-TBAPy in removing toxic GP intermediates from the solution.

## Photooxidation pathways of GP

Despite the absence of open metal sites in M$^{3+}$-TBAPy MOFs, their nanoconfined pore spaces facilitate the adsorption and complexation of GP molecules through electrostatic van der Walls interactions[73]. Upon light irradiation under oxic conditions, the MOFs become activated, leading to the generation of e$^-_{CB}$ and h$^+_{VB}$. These species can react with the solvent medium, such as water or O$_2$, present in the reaction, resulting in the formation of ROS such as •OH, O$_2^{\bullet-}$ and $^1$O$_2$, which are known to be the primary agents in oxidative reactions[91,92]. The specific ROS present in the reaction system determines targeted bond cleavage sites in GP, leading to the formation of different products. The electrophilic nature of the C–P bond in GP makes it susceptible to nucleophilic attack by the MOFs[93], as demonstrated in previous studies with semiconductors like MnO$_2$[94]. However, in our reactions, neither sarcosine nor orthophosphate, resulting from C–P bond cleavage, was observed. This result suggests that abiotic degradation of GP does not necessarily proceed through this pathway, possibly due to the suppression of the C–P Lyase pathway in the reaction. Based on the observed products, it is plausible that the GP degradation proceeds through the cleavage of its two C–N bonds. Our recent study highlighted the direct formation of glycine from GP using MOF-derived semiconductors, resulting from β C–N bond cleavage, as well as the formation of AMPA from α C–N bond cleavage, both of which are consistent with the behavior seen with our M$^{3+}$-TBAPy MOFs[54]. This finding indicates that the degradation of GP by the M$^{3+}$-TBAPy MOFs involves the cleavage of C–N bonds, resulting in the formation of mixed products, including glycine, formic acid, acetic acid and AMPA.

Interestingly, among all the M$^{3+}$-TBAPy MOFs, only Sc-TBAPy demonstrates selectivity toward the formation of glycine and formic acid, while Fe-TBAPy, Al-TBAPy and Y-TBAPy yielded AMPA as one of the products. Fe-TBAPy's lower activity for GP oxidation is rationalizable by its less positive VB potential (Supplementary Fig. 2), resulting in a diminished GP oxidative ability compared to the other MOFs. Although Sc-TBAPy, Y-TBAPy, and Al-TBAPy exhibit similar ground-state electronic structures based on their density of states (DOS), their reactivities toward GP oxidation differ significantly, confirming that the metal identity in the MOF plays a crucial role. The observed behavior of Sc-TBAPy reveals a dominant pathway via a β C–N bond cleavage to form glycine, in difference from the other MOFs. Analysis using Inductively Coupled Plasma Optical Emission spectroscopy (ICP-OES) was performed to identify metal leaching during the GP photodegradation reaction with Sc-TBAPy. After an 8 h reaction, the analysis revealed the presence of 1.6 ppm of Sc$^{3+}$ in the solution, indicating only a 0.42% degradation of the Sc-TBAPy (Supplementary Table 5). These results confirm the stability of Sc-TBAPy after long-term operations in aqueous solution in the presence of ROS. The minimal metal leaching observed is likely to have no or minor impact on the GP degradation process. Based on these correlated experimental and computational findings, we can thus attribute Sc-TBAPy's superior catalytic activity to (i) the strong interaction between the metal and TBAPy$^{4-}$, which enhances charge transfer and separation through LMCT, (ii) the formation rate and lifetime of favorable ROS species such as $^1$O$_2$, which sufficiently drives the reaction for Sc-TBAPy, and (iii) the induced electronic effects resulting from van der Waals interactions and the spatial orientation of GP within the nanoconfined spaces. These key effects contribute to the increased adsorption and efficient degradation of GP.

## Impact of ROS on GP degradation

Besides the above-mentioned experiments, we also performed oxidation reactions under anoxic conditions and with deuterated water (D$_2$O) to investigate the role of active ROS in GP degradation using Sc-TBAPy and Al-TBAPy (Table 1). Previous studies showed that anoxic conditions lead to the formation of aqueous e$^-_{CB}$, which can reduce O$_2$

**Table 1 | ROS formation for the GP degradation under different reaction conditions**

| Reaction Conditions | ROS formation | | Product formation | |
|---|---|---|---|---|
| | Sc-TBAPy | Al-TBAPy | Sc-TBAPy | Al-TBAPy |
| **Oxic ($H_2O$)** | $^1O_2 > \cdot OH > O_2^{\cdot-}$ | $^1O_2 > O_2^{\cdot-} > \cdot OH$ | Glycine, formic acid | GP, AMPA, formic acid, acetic acid |
| **Anoxic ($H_2O$)** | $O_2^{\cdot-} > {}^1O_2 > \cdot OH$ | $O_2^{\cdot-} > {}^1O_2 > \cdot OH$ | Glycine, formic acid | GP, formic acid, acetic acid |
| **Oxic ($D_2O$)** | $^1O_2$ has a longer lifetime | | Glycine, formic acid | AMPA, formic acid, acetic acid |

into $O_2^{\cdot-}$ radicals and contribute to the overall photocatalytic activity[53]. Analysis of the $^1H$ NMR spectra for Al-TBAPy under anoxic conditions revealed the absence of AMPA, with formic acid and acetic acid as dominant products, similar to the observed products under oxic conditions but now with the absence of AMPA (Supplementary Fig. 26). Integration of the $^1H$ NMR peaks for Al-TBAPy showed higher product yields, with ratios of 27:1 for acetic acid and 6:1 for formic acid under anoxic vs. oxic conditions. In the case of Sc-TBAPy, the $^1H$ NMR analysis showed that formic acid and glycine are the dominant products, as previously observed. Integration of $^1H$ NMR peaks under anoxic vs. oxic conditions showed a ratio of 4:1 for glycine and 3:1 formic acid. Based on our findings, Al-TBAPy performs better under anoxic conditions, possibly due to the contribution of aqueous $e_{CB}^-$ which enhances the formation of $O_2^{\cdot-}$.

To investigate the role of $^1O_2$ species, the oxidation reaction was conducted in $D_2O$ as a solvent exchange for $H_2O$. Previous studies suggested that $^1O_2$ has a longer lifetime in $D_2O$, enhancing the removal of pollutants in water[53,95,96]. Using Al-TBAPy as the catalyst, there was a complete conversion of GP in $D_2O$, whereas 1.67% remained in solution when $H_2O$ was used. Integration of the $^1H$ NMR peaks in $D_2O$ vs. $H_2O$ showed the increased formation of AMPA, formic acid and acetic acid in $D_2O$ with ratios 3:1 for AMPA, 2:1 for formic acid, and 6:1 for acetic acid. Similarly, Sc-TBAPy showed increased product yields in $D_2O$ compared to $H_2O$, with integration of the $^1H$ NMR peaks indicating ratios of 2.6:1 for formic acid and 1.3:1 for glycine (Supplementary Fig. 27). These observations infer that $^1O_2$ species contributes to the overall oxidation of GP in $D_2O$ with both Sc-TBAPy and Al-TBAPy. To further validate these findings, electron spin resonance (ESR) spectroscopy was performed on Sc-TBAPy and Al-TBAPy to monitor ROS formation. Under dark conditions, no ESR signal was observed for Sc-TBAPy and Al-TBAPy, indicating the absence of ROS formation. Upon light irradiation, $\cdot OH$, $O_2^{\cdot-}$ and $^1O_2$ were detected. The pertinent signal intensities in the $\cdot OH$, $O_2^{\cdot-}$ and $^1O_2$ spectra increased with longer irradiation from 1 to 2 min (Supplementary Figs. 28, 29). Analysis of the intensity vs. time plots for the $\cdot OH$, $O_2^{\cdot-}$ and $^1O_2$ signals (Fig. 6) revealed that Sc-TBAPy exhibits higher intensities for all detected ROS than Al-TBAPy, indicating a higher rate of ROS formation in Sc-TBAPy. Specifically, the formation yield of $^1O_2$ was the highest in both MOFs, while $O_2^{\cdot-}$ and $\cdot OH$ were observed at lower levels in Sc-TBAPy and Al-TBAPy. Based on ESR analysis, we concluded that the formation of $^1O_2$ favors the GP degradation, particularly with Sc-TBAPy, and likely contributes to the selective oxidation of GP. The non-selective catalysis observed with Al-TBAPy is likely due to the lower rate of $^1O_2$ formation and presence of $O_2^{\cdot-}$ and $\cdot OH$. The role of the active ROS present in the oxidation of GP with Sc-TBAPy was further probed through the utilization of ROS scavengers. These scavengers, namely isopropanol (IPA), p-benzoquinone (p-BQ), and sodium azide ($NaN_3$), were employed to quench $\cdot OH$ radical, $O_2^{\cdot-}$ radical and $^1O_2$ respectively. Observations derived from the experimental results indicated the complete disappearance of GP from solution, accompanied by the formation of glycine and formic acid. The relative concentrations were determined as 1:2 for p-BQ, 1:1 for IPA, and 1:2 for $NaN_3$ based on the integration of the corresponding peaks in the $^1H$ NMR spectra (Supplementary Fig. 30). These findings strongly suggest the substantial contribution of all three ROS to the oxidation of GP. Overall, the observed ROS in the reaction plays an active role in the degradation of

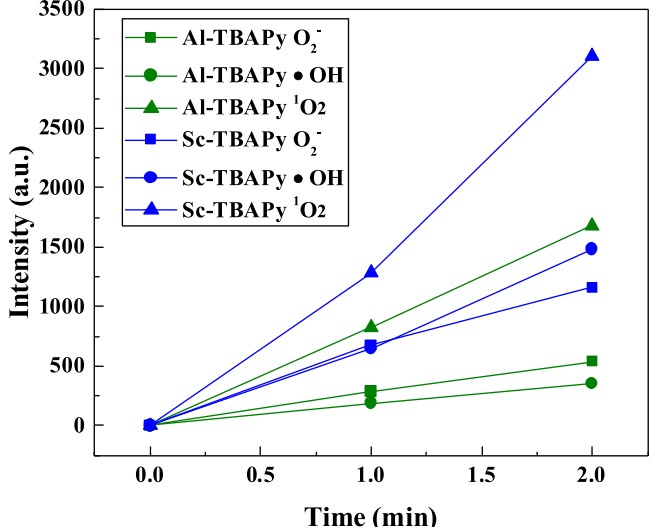

**Fig. 6 | Comparison of ROS formation by Sc-TBAPy (blue) and Al-TBAPy (green) using ESR spectroscopy.** The intensity vs. time plots of $O_2^{\cdot-}$, $\cdot OH$, and $^1O_2$ reveal that Sc-TBAPy exhibits higher ROS formation compared to Al-TBAPy upon light irradiation.

GP, but the rate of formation, lifetime and the individual contribution of each ROS species can influence and vary the degradation pathway of GP (Fig. 7). Comparing Sc-TBAPy with benchmark materials for GP oxidation in the literature (Supplementary Table 6) reveals that our Sc-TBAPy performs favorably in both GP adsorption and degradation. It demonstrates selective catalytic activity in the degradation of GP, yielding benign products and showcasing its practical utility for wastewater treatment.

In this study, we investigated the performance of four isostructural pyrene-based MOFs with different metal nodes for the adsorption and photodegradation of GP. Our results revealed that the choice of metal nodes can significantly influence the applicability of MOFs for GP remediation. While all MOFs exhibit the adsorption and GP degradation capabilities, Sc-TBAPy outperforms the other $M^{3+}$-TBAPy MOFs. Remarkably, despite having the lowest BET surface area among the examined materials, Sc-TBAPy demonstrates the fastest adsorption of GP, highlighting the importance of electrostatic interactions in the adsorption process and tunability of MOFs by varying the metal nodes. Furthermore, Sc-TBAPy exhibits superior photocatalytic activity in the reduction of $H^+$ to produce $H_2$ and oxidation of GP, indicating its enhanced charge carrier generation ability, nicely corroborated by fs-TA spectral analysis on the transient CT band formation and excimer decay dynamics on ultrafast timescales. Last but not least, Sc-TBAPy exhibits selectivity in the degradation of GP, avoiding the formation of toxic AMPA, which is a commonly observed undesirable product in oxidative processes. Our experimental investigations, including ESR, depleted oxygen, and $D_2O$ experiments, unveil that $^1O_2$ plays a pivotal role in the degradation performance of GP by Sc-TBAPy, potentially contributing to its selectivity. These fundamental insights underscore the potential of MOFs for the removal of GP from aqueous systems and highlight the ability to tune their

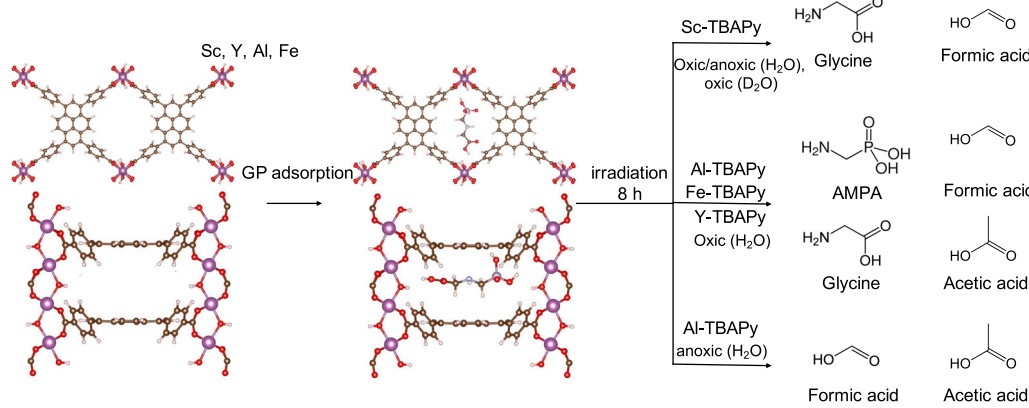

**Fig. 7 | Product formation by the M³⁺-TBAPy MOFs under different conditions.** The branched pathways highlight the impact of ROS on GP degradation.

structures for selectively breaking down GP and other water contaminants. This work is envisioned to serve as a foundation for the rational development and design of more efficient, selective, and high-capacity materials to effectively remove organic pollutants from water supplies.

## Methods
All chemicals and solvents used in this work were purchased from commercial sources and used without further purification.

### Synthesis of Sc-TBAPy MOF
A combination of scandium (III) nitrate hydrate (20 mg, 0.080 mmol; 99.9% trace metals basis), 1,3,6,8-tetrakis(*p*-benzoic acid)pyrene (H₄TBAPy) (10 mg, 0.014 mmol; purity >95%), and 3 drops of concentrated hydrochloric acid was dissolved in 5 mL of dimethylformamide (DMF; Percent Purity, ≥99.8%) along with 0.5 mL of deionized water (DI-water). This mixture was stirred until it became a homogeneous solution. It was then heated within a scintillation vial at a temperature of 120 °C for 48 h. Subsequently, the mixture was cooled to room temperature and the product was filtered and washed with DMF (3×). The Sc-TBAPy crystals were immersed in methanol (MeOH) for 24 h to replace any solvent molecules within the structure. Finally, the Sc-TBAPy crystals were left to air-dry overnight[77].

### Synthesis of Y-TBAPy MOF
Y-TBAPy was prepared following a method akin to the synthesis of Sc-TBAPy, with the exception being the utilization of yttrium (III) nitrate hexahydrate (20 mg, 0.052 mmol; 99.8% trace metals basis) in place of scandium (III) nitrate.

### Synthesis of Al-TBAPy MOF
Al-TBAPy was prepared following a method reminiscent of the synthesis of Sc-TBAPy. The synthetic difference lies in the use of aluminum (III) nitrate nonahydrate (20 mg, 0.053 mmol; 99.8% trace metals basis) instead of scandium (III) nitrate hydrate, and the adjustment of the hydrothermal reaction conditions to 120 °C for a duration of 12 h.

### Synthesis of Fe-TBAPy MOF
A mixture of iron (III) nitrate nonahydrate (20 mg, 0.050 mmol; 99.95% trace metals basis), H₄TBAPy (10 mg, 0.014 mmol), and 3 drops of hydrochloric acid was dissolved in 5 mL of DMF and 0.5 mL of DI-water. The mixture was stirred to create a homogeneous solution, which was then subjected to heating in a scintillation vial at 120 °C for a duration of 48 h. The vial was promptly removed from the oven immediately after the completion of the reaction. The product was filtered and washed with DMF (3×). The Fe-TBAPy crystals were immersed in methanol (MeOH) for 24 h to facilitate the replacement of any solvent

molecules within the structure. Ultimately, the Fe-TBAPy crystals were left to air-dry overnight.

### Synthesis of Ni₂P nanoparticles (NPs)
Ni-BTC MOF was first synthesized by mixing nickel (II) nitrate hexahydrate (Ni(NO₃)₂·6H₂O) (109 mg, 0.37 mmol; 99.99% trace metals basis) and trimesic acid (H₃BTC) (43.7 mg, 0.21 mmol; purity 95%) in 6 mL methanol. The mixture was then heated at 150 °C for 24 h. The as-made Ni-BTC was washed with methanol and dried at 60 °C. For synthesizing Ni₂P NPs, 20 mg of the as-made Ni-BTC was mixed with 60 mg of sodium hypophosphite monohydrate (NaH₂PO₂·H₂O; purity >99%) and heated to 275 °C for 2 h in a furnace. The as-synthesized Ni₂P was washed with water and ethanol and dried at 75 °C[62].

### Photocatalytic hydrogen evolution half reaction
The hydrogen evolution reaction for all four MOFs was performed in a solution of acetonitrile and water, under the same condition for ease of comparison. For each reaction, 16.6 mL acetonitrile (purity >99.9%), 100 μL TEA (purity >99.5%), and 298 μL of DI water were transferred into a 6-dram Pyrex glass vial acting as the reactor. 17 mg of the MOF and 1.8 mg Ni₂P were added to the reactor which was tightly capped with a rubber septum. The solution was stirred under argon for 20 min to purge out any dissolved oxygen and then irradiated under constant stirring with a 300-W Xe lamp which was supported by a 360 nm cut-off filter. To measure the hydrogen produced, 0.2 mL of the gaseous product was abstracted from the headspace and analyzed using gas chromatography.

### Photocatalytic oxidation half reaction
A 1.5 mM GP solution (GP: analytical standard) was prepared, and 3 mL solution was transferred into a 1-dram reaction vial with 10–15 mg of MOFs. The solution was then irradiated by a 300-W Xe lamp (which was supported by a 360 nm cut-off filter) under oxic conditions for 8 h with constant stirring, using a stir bar to keep the solution well mixed with the powdered catalyst evenly dispersed.

## Data availability
The data that supports the findings of the study are included in the main text and supplementary information files. Raw data can be obtained from the corresponding author upon request.

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

## Acknowledgements

K.C.S. thanks the Department of Chemistry at Oregon State University (OSU) for support through start-up funding. N-.C.C., E.N.M., and J.M.L. acknowledge support from the Renewable Energy Award, OSU Department of Chemistry for the summer fellowship, and the College of Science Industry Partnership award, respectively. C.F. thanks the U.S. National Science Foundation (NSF) grant CHE-2003550, and OSU College of Science SciRIS-ii award (2022–2023) for financial support. L.S.L. acknowledges the OSU Department of Chemistry for summer fellowships (2022 and 2023).

## Author contributions

K.C.S. conceived the project and designed the experiments together with N.-C.C., J.M.L., E.N.M., and M.T.N. N.-C.C., J.M.L., and E.N.M. led the experimental work (synthesis and characterization), performed the GP degradation experiments and interpreted the data. L.S.L., T.D.K., C.C., and C.F. performed the fs transient absorption experiments and analyzed the data. C.W. and P. H.-Y.C. led the theoretical calculations described in this work. H.H. conducted the ESR experiments and analyzed the data. T.C.G. performed the EIS experiments. All authors contributed to the writing and editing of the manuscript.

## Competing interests

The authors declare no competing interests.
