## [Peer Review File · Nature Communications]

Elucidation of the role of metals in the adsorption and photodegradation of herbicides by metal-organic frameworksREVIEWER COMMENTS

Reviewer #1 (Remarks to the Author):

This study investigated the role of metals in a MOF structure on the adsorption and photocatalytic degradation of a pesticide. The authors investigated the effect of four metals on the photophysical properties of isostructural MOF and their effectiveness for adsorption and photocatalytic degradation of a pesticide. The topic is of interest and the study is overall solid. The main drawbacks of this manuscript in my opinion are (1) the writing is lengthy, and (2) the article did not discuss at all any potential impact of water constituents on the performance of the concerned MOF. Some specific comments of this manuscript are shown below:

Introduction is verbose. The authors could significantly shorten the context without losing any contents. For example, the first paragraph should be condensed into 3 sentences.

In page 3, the first paragraph, the authors stated that “the primary disadvantage of ...from environmental water”. what do you mean that adsorption results in the accumulation of adsorbents? this sentence does not make sense to me.

Why does acidic condition benefit GP removal? I suppose you are referring to the adsorption process here.

In the second paragraph of this same page, the explanation of photocatalysis is unnecessary because the readers of the natural communication should have some preliminary understanding on the photocatalytic process.

On page 5, what is NSTA?

Page 7, here you said Sc-MOF has the highest charge transfer resistance property, in Figure 2, you described it has the highest charge transfer. This appears not consistent to me.

Page 8, you do not need to explain what bandgap is. Please delete the sentence

Page 8: you need to explain how XPS is used to calculate the valance band.

Page 9: wondering what the excimer is?

Page 9 and 10: the last sentence of page 9 is not clear to me. The shorter emission lifetime of what?

Page 13: define DMF please

Page 13: I did not see a control that measures the metabolites of GP in the dark. How could you be fully sure that adsorption is the only process that drives the removal of GP from the solution?

Page 15, does the degradation kinetics differ in oxic vs anoxic conditions?

Reviewer #2 (Remarks to the Author):

In this work, the author synthesized a series of metal organic frameworks and apply them for photocatalytic oxidation of pollutants in water. The author analysed the species of degradation pathways during the reactions as well as physicochemical properties of the MOF materials. My major consider is that MOF materials are normally not good candidate for its instability in water, high cost, and poor

resistant to ROS. The proposed elements are typically heavy metals which would lead to secondary contaminations. The catalyst was not compared with the benchmark materials in the literature. I could not see any significant advances in materials development or system innovation. Therefore, I could not recommend its publication in Nature Commun.

Other comments include:

The authors need to evaluate the stability of the materials after long-term operations. Under strong oxidizing environment, most MOF with organic ligand could not survive.

Also evaluated the metal leaching after the reaction, and these metal ions may also have impact on the degradation process.

The ROS were not related with the degradation pathways. The MS analyses of the byproduct normally provide evidence for the corresponding ROS.

Gentle reminder the EPR peak intensity could NOT advise the corresponding ROS pollution. The author need to use organic probes and kinetic calculations to perform the analysis.

The evolution of different ROS should be revealed to gain more mechanistic insights.

Reviewer #3 (Remarks to the Author):

In this paper entitled “Discovering a herbicide terminator: the role of metals in elimination glyphosate by Metal-Organic Frameworks” authors reported a 3 modified TBAPy MOF based on the Sc-TBAPy and studied their use in GP elimination from water (adsorption and photocatalysis). Although the important use of GP in our fields, it is an important contaminant of our environments. The authors demonstrated for the first time that MOFs can be used as catalyst in the degradation of GP. The text is clear and easy to read. However, I believe that the claims are high, and some more data and experiments are needed.

Therefore, I recommend a major revision. It will be great to accept the manuscript after the demanded information, experiments, and changes. I strongly believe that this is a great work, but it can be superior.

1. In my opinion, the introduction is quite extensive and impressive. But only a work is mentioned on the use of MOFs in GP elimination of water. Other articles should be also mentioned:

<https://doi.org/10.1016/j.jpcc.2021.110403>, <https://doi.org/10.1021/acsomega.8b00921> and

<https://doi.org/10.1002/adfm.202213862>

2. Probably authors should comment in the abstract that the degradation of GP is studied in water media.

What is the initial concentration of GP in water in the elimination processes (adsorption and photodegradation)? As I see in the oxidation half reaction 1.5 mM of GP solution is used. Why authors have selected this concentration? Which is the GP concentration found in contaminated waters? The selected GP concentration in the elimination test should be in accordance with the normally concentration found in waters.

Authors assess the stability of Sc-TBAPy after the photocatalytic process by different techniques: FTIR, DXRP and BET surface measurements. Interestingly the BET surface increased once the Sc-TBAPy was

used in GP in UV-light and washed DMF. Probably some degradation of the framework occurs during this process (from 793 to 884 mg/g). Considering that the final application is water remediation, authors should check if there is some linker or metal on the final water solution after the GP degradation process.

Further, the recyclability of Sc-TBAPy in the photocatalytic tests should be done in order to assess this point. Is it Sc-TBAPy stable? How many cycles can we use Sc-TBAPy to degrade?

During the introduction authors mentioned some problems of the actually studied methods: 1) SELECTIVITY “there remains an essential need to develop selective photocatalysts”, 2) REUSABILITY “Post-treatment recovery of adsorbents remains challenging, which limits the ability to easily reuse these materials for GP capture”, and LAB SCALE 3) “most studies have been conducted at laboratory scales using controlled conditions, while high efficiency of GP removal required acidic conditions”

In this work, there is no selectivity test, the reusability of the photocatalyst is not included and the test have been performed at lab scale. So, when compared the state of the art in the use of different materials in GP elimination from water and the final results I cannot see any reasons about the selection of MOFs in GP elimination. May be the introduction is highly ambitious and should be modified.

Response to the Reviewers

Reviewer 1:

This study investigated the role of metals in a MOF structure on the adsorption and photocatalytic degradation of a pesticide. The authors investigated the effect of four metals on the photophysical properties of isostructural MOF and their effectiveness for adsorption and photocatalytic degradation of a pesticide. The topic is of interest and the study is overall solid. The main drawbacks of this manuscript in my opinion are (1) the writing is lengthy, and (2) the article did not discuss at all any potential impact of water constituents on the performance of the concerned MOF. Some specific comments of this manuscript are shown below:

1-1. Introduction is verbose. The authors could significantly shorten the context without losing any contents. For example, the first paragraph should be condensed into 3 sentences.

Response. We appreciate the feedback from reviewer 1, and as per the suggestions, we have considerably shortened and improved our introduction.

1-2. In page 3, the first paragraph, the authors stated that “the primary disadvantage of ...from environmental water”. What do you mean that adsorption results in the accumulation of adsorbents? this sentence does not make sense to me.

Response. We thank reviewer 1 for the comment. We acknowledge that the previous statement was unclear. Our intention was to convey that as porous materials accumulate guest molecules, their efficiency to absorb new molecules decreases, due to accumulated adsorbents building up on the materials surface. To provide clarity and avoid confusion, we have revised the statement as follows: *“The primary disadvantage of these materials is that recovery of adsorbates is challenging, and accumulation of adsorbates over time reduces their efficiency.”³⁰*

1-3. Why does acidic condition benefit GP removal? I suppose you are referring to the adsorption process here.

Response. We thank reviewer 1 for the comment. In acidic conditions, porous materials exhibit higher efficiency in capturing glyphosate. In basic conditions, the net negative charge of the material and GP increase, leading to increased electrostatic repulsion, which hinders the adsorption process. We have revised this sentence in our manuscript to make this statement clear. The sentence reads: *“Moreover, high efficiency of GP removal has often required acidic conditions, as the net charge of porous materials and GP become more negative with increased pH, thereby increasing electrostatic repulsion and hindering adsorption.”^{30,38}*

1-4. In the second paragraph of this same page, the explanation of photocatalysis is unnecessary because the readers of the natural communication should have some preliminary understanding on the photocatalytic process.

Response. We appreciate reviewer 1 for the suggestion, and we have removed the explanation of photocatalysis. That said, we chose to keep the description of ROS generation, as the variety of ROS in our system is a critical element in our narrative.

2. On page 5, what is NSTA?

Response. We thank reviewer 1 for the comment. NSTA is the abbreviation of nitrogen and sulfur-doped TiO₂ anatase, which is derived from the metal-organic framework (MOF) MIL-125-NH₂. More details of NSTA can be found the reference in the manuscript: Musa, E. N. *et al.* Two Birds, One

Stone: Coupling Hydrogen Production with Herbicide Degradation over Metal–Organic Framework-Derived Titanium Dioxide *ACS Catal.* **13**, 3710-3722 (2023).

3. Page 7, here you said Sc-MOF has the highest charge transfer resistance property, in Figure 2, you described it has the highest charge transfer. This appears not consistent to me.

Response. We thank reviewer 1 for catching this. Sc-TBAPy has the largest semicircle radius which indicates high charge transfer resistance. We revised the caption in Figure 2. The caption reads: “*Figure 2. Solid-state characterization of the four M^{3+} -TBAPy MOFs. a. Powder X-Ray diffraction (PXRD) patterns. All PXRD patterns of MOFs match the simulated M^{3+} -TBAPy PXRD pattern (black), indicating that they are isostructural and can be synthesized as phase pure. b. Type I N₂ isotherms collected at 77 K and 1 bar reveal that all MOFs are microporous. c. EIS Nyquist plots reveal that Sc-TBAPy (blue) has the largest semicircle radius, representing the highest charge transfer resistance. d. Kubelka–Munk transformation diffuse reflectance spectra collected from 200 to 800 nm.*”

4. Page 8, you do not need to explain what bandgap is. Please delete the sentence

Response. We thank reviewer 1 for the comment. We removed the sentence explaining bandgap. The paragraph in the manuscript “Photophysical Properties section” on page. 8 reads: “...*Energy bands, including the VB, CB and bandgap (Eg), are crucial properties of photocatalytic materials. The bandgap, representing the energy required to excite an electron from the VB to the CB, was calculated from the diffuse reflectance spectroscopy data. ...*”

5. Page 8: you need to explain how XPS is used to calculate the valance band.

Response. We thank reviewer 1 for the comment. We provide more details explaining how the valance band values of all MOFs were calculated from XPS data. The sentence in the manuscript “Photophysical Properties section” reads: “... *XPS analysis with linear extrapolation method provided insights into the experimental VB for all MOFs,⁸⁰⁻⁸¹ and the CB of each MOF was determined by subtracting the bandgap energy from the VB potential energy (Figures S2, S3, S4, and Table S2). ...*”. The paragraph in the supporting information “X-ray photoelectron spectroscopy (XPS)” section reads: “...*XPS experiments were measured on as-loaded samples at room temperature. The valence band value (or valence band maximum) of all MOFs was determined by using a linear extrapolation method. By linear fitting the leading edge of the valence band and linearly fitting the flat energy distribution, the intersection of these two lines is the experimental VBM value (Figure S4).*^{3,4}”

6. Page 9: wondering what the excimer is?

Response. We thank reviewer 1 for the comment. The excimer mentioned in the manuscript is one of the emissive species that could form in MOFs with densely packed linkers in highly ordered arrays. These emissive species may occur within the MOFs via interchromophoric interactions. We have revised the paragraph to provide more explanation and prevent confusion. The paragraph in manuscript reads: “...*This late SE band, supported by the TA spectra of the ligand (Figure S9), is attributed to the formation of emissive excimers. Excimers could form in MOFs with densely packed linkers in highly ordered arrays. These emissive species may occur within the MOFs via interchromophoric interactions.^{8,13,14} Although the rise of the excimer SE band in the MOFs is considerably slower compared to the aggregated state of the ligand (Figure S9d, ~300 ps retrieved for the suspended ligand in acetonitrile (ACN)) and exhibits significant variation among the MOFs (Figures S6a, S7, and S8)....*” and “...*Detailed analysis and additional discussion regarding fs-TA*

can be found in the Supporting information. This important finding explains the minimal broadening of fluorescence observed in the MOFs since nonradiative pathways dominate over fluorescence as the primary relaxation pathway for MOF excimers.”

7. Page 9 and 10: the last sentence of page 9 is not clear to me. The shorter emission lifetime of what?

Response. We thank review 1 for the comment. The shorter emission lifetime is corresponding to the excimer. We have revised the sentence to clarify this and prevent any confusion. The sentence in the manuscript on page 9 reads: *“Meanwhile, the shorter emission lifetime of the excimer indicates an increased nonradiative decay of H₄TBAPy ligands. This is attributed to greater pyrene-pyrene distances in the MOF as well as increased solvent interactions, thus promoting the quenching of the excimer state.^{84”}*

8. Page 13: define DMF please

Response. We thank reviewer 1 for their comment, and as per the suggestion, we defined DMF and the revised sentence reads: *“To verify that GP was not merely adsorbed within the pores of Sc-TBAPy, the MOF was recovered after the photooxidation reaction and washed with dimethylformamide (DMF) to facilitate the removal of any GP within its pores....”*.

9. Page 13: I did not see a control that measures the metabolites of GP in the dark. How could you be fully sure that adsorption is the only process that drives the removal of GP from the solution?

Response. We thank reviewer 1 for the comments and the question raised. To address this, we performed experiments with glycine and AMPA, two primary metabolites of GP observed in our reactions after photocatalysis, using Sc-TBAPy under dark conditions. The results obtained from the ¹H-NMR analysis revealed that glycine remained in solution post-reaction, whereas no trace of AMPA was detected. The different behavior observed with AMPA and glycine indicates the selective nature of the Sc-TBAPy towards the removal of toxic intermediates, such as AMPA (with Lewis basic phosphonate groups), from solution. This result has been added in the manuscript and our revised text reads: *“...In addition, adsorption experiments for AMPA and glycine, two main metabolites of GP, were conducted under dark conditions using Sc-TBAPy. Analysis of the ¹H NMR spectra revealed that a significant amount of glycine, approximately 99.4%, remained in the solution after the reaction (Figure S25a). However, no traces of AMPA were detected in the solution post-reaction (Figure S25b). Our rationale for this different sorption behavior between AMPA and glycine is attributed to the phosphonic acid group present in AMPA, akin to that in GP, which enables binding to the MOF. In contrast, such binding is minimal in glycine due to the absence of the phosphonic acid group. These results further support the selective nature of Sc-TBAPy in removing toxic GP intermediates from the solution.”*

Regarding the comment *“How could you be fully sure that adsorption is the only process that drives the removal of GP from the solution?”* Our study shows evidence for the mechanism of adsorption and degradation of GP, indicating that both adsorption and degradation drives the process of GP removal from the solution. As we describe in our manuscript, the removal of GP from the solution was investigated with the Sc-TBAPy under dark and light conditions, and our findings reads: *“...The rapid removal of GP by Sc-TBAPy without light prompted us to delve deeper into whether fast GP removal by Sc-TBAPy with light was due to adsorption or photodegradation. To investigate this, we first performed the reaction in the dark for three consecutive cycles. Analysis of the ¹H NMR data obtained after the first cycle revealed no trace of GP in solution (Figure S18a). However, after the second cycle, GP remained in the solution. Additionally, upon completion of the third cycle, the ¹H*

NMR results indicated a higher concentration of GP in the solution (Figure S18a). This unexpected finding led us to propose that the active sites of Sc-TBAPy may have become saturated, inhibiting further GP uptake. Alternatively, when the reaction was conducted under light for three consecutive cycles, ¹H NMR analysis indicated the absence of GP in the solution (Figure S18b). The reaction under light was subsequently performed over three consecutive cycles giving glycine and formic acid, with no sign of GP in solution as confirmed from the ¹H NMR analysis (Figure S18b). The absence of GP in solution indicates its complete degradation and the ability of the Sc-TBAPy MOF to photodegrade GP over three cycles.”

“...Therefore, we propose that GP is initially captured and retained within the pores of the Sc-TBAPy, ultimately reaching a saturation point in the absence of light. However, under light irradiation, the continuous generation of ROS facilitates the attack on GP, causing the cleavage of relevant bonds and the release of degradation products from the MOF’s pores. Consequently, the active sites and pores of Sc-TBAPy are regenerated, enabling further adsorption and catalysis, as conformed from the recycling experiments (Figure S19).”

These observations indicate that the mechanism of both adsorption and photodegradation drives the process of removal of GP from solution with Sc-TBAPy.

10. Page 15, does the degradation kinetics differ in oxic vs anoxic conditions?

Response. We thank reviewer 1 for raising a valid question. The results from the GP oxidation experiments suggest that the degradation kinetics for product formation differ depending on the conditions applied. Specifically, with Al-TBAPy, a higher product yield was measured under anoxic compared to oxic conditions. However, using Sc-TBAPy, a greater yield in product formation was observed under oxic conditions compared to anoxic conditions. This difference is attributed to the presence of aqueous electrons in the reaction medium. This part is revised in our manuscript, and it reads: *“Integration of the ¹H NMR peaks for Al-TBAPy showed higher product yields, with ratios of 27:1 for acetic acid and 6:1 for formic acid under anoxic vs. oxic conditions. In the case of Sc-TBAPy, the ¹H NMR analysis showed that formic acid and glycine are the dominant products, as previously observed. Integration of ¹H NMR peaks under oxic vs. anoxic conditions showed a ratio of 4:1 for glycine and 3:1 formic acid. Based on our findings, Al-TBAPy performs better under anoxic conditions, possibly due to the contribution of aqueous e⁻_{CB} which enhances the formation of O₂⁻.”*

Reviewer 2:

1. In this work, the author synthesized a series of metal organic frameworks and apply them for photocatalytic oxidation of pollutants in water. The author analysed the species of degradation pathways during the reactions as well as physicochemical properties of the MOF materials. My major consider is that MOF materials are normally not good candidate for its instability in water, high cost, and poor resistant to ROS. The proposed elements are typically heavy metals which would lead to secondary contaminations. The catalyst was not compared with the benchmark materials in the literature. I could not see any significant advances in materials development or system innovation. Therefore, I could not recommend its publication in Nature Commun.

Response. We thank reviewer 2 for their evaluation of our work and insightful comments. Indeed, some of the components used in synthesizing MOFs are considered heavy metals, contributing to the potential expense of synthesizing these MOFs. However, it is essential to highlight that our study provides fundamental insights that are vital for the design and development of more efficient, selective, and high-capacity materials aimed at effectively removing organic pollutants from water. Moreover, we conducted an examination of the stability of the Sc-TBAPy MOF, affirming that it retained its crystallinity after the photodegradation reaction. Further investigation using Inductively Coupled Plasma Optical Emission spectroscopy (ICP-OES) to analyze the GP solution after an 8 hr reaction with Sc-TBAPy revealed 1.6 ppm Sc³⁺ in the solution, signifying only a 0.42% degradation of Sc-TBAPy after the GP photodegradation reaction. Therefore, Sc-TBAPy MOF is considered stable either in water, during photocatalytic GP degradation reactions, and in the presence of ROS. There is a minimal contamination (0.42% degradation of Sc-TBAPy MOF) following the GP photodegradation reaction. We have revised the results and discussion in the manuscript as well as the instrumentation method of the ICP-OES section, as detailed in the supporting information. The revised paragraph in manuscript reads: “...Analysis using Inductively Coupled Plasma Optical Emission spectroscopy (ICP-OES) was performed to identify metal leaching during the GP photodegradation reaction with Sc-TBAPy. After an 8 hr reaction, the analysis revealed the presence of 1.6 ppm of Sc³⁺ in the solution, indicating only a 0.42% degradation of the Sc-TBAPy (Table S5). These results confirm the stability of Sc-TBAPy after long-term operations in aqueous solution in the presence of ROS. The minimal metal leaching observed is likely to have no or minor impact on the GP degradation process....” .

Addressing the reviewer’s comment, “the catalyst was not compared with the benchmark materials in the literature I could not see any significant advances in materials development or system innovation. Therefore, I could not recommend its publication in Nature Commun.” A comparative analysis of our MOF with the benchmark materials for GP oxidation in the literature has been presented below and in Table S6 of the SI. The data demonstrates that our Sc-TBAPy performs favorably in both the adsorption and degradation of GP, with a particular emphasis on generating benign degradation products. This comparison underscores the practical application of our MOF in wastewater treatment.

Other comments include:

2. The authors need to evaluate the stability of the materials after long-term operations. Under strong oxidizing environment, most MOF with organic ligand could not survive.

Response. We thank reviewer 2 for their valid comment. We agree that the strong oxidizing environment significantly influences the stability of MOFs. Consequently, we conducted the GP

oxidation experiment in both natural river water and basic solutions with a pH of 10.7. The results indicate that Sc-TBAPy not only effectively degrades GP but also maintains its crystallinity in both natural river water and basic conditions. The results of our findings are now shared in our revised manuscript and SI: “...To investigate the impact of Sc-TBAPy in the degradation of GP in a natural environment, a 1.5 mM GP solution was prepared using sampled river water. Additionally, another GP solution with the same concentration was prepared, and its pH was adjusted to 10.7 using 1M NaOH to evaluate the activity of the MOF in an oxidizing environment. Subsequently, the reactions were performed. Results from ^1H NMR analysis revealed complete degradation of GP in the solution, leading to the formation of glycine and formic acid (Figure S23). To verify the stability Sc-TBAPy under basic conditions, its PXRD pattern was collected. The results demonstrated that the recovered Sc-TBAPy remained stable after exposure to irradiation in a pH 10.7 GP solution, indicating the superior activity of the MOF in both acidic and basic environments (Figure S24).”

Additionally, ICP-OES analysis was conducted on the solution following an 8 hr GP photodegradation reaction with Sc-TBAPy. The result revealed the presence of 1.6 ppm of Sc^{3+} in the solution, indicating only a 0.42% Sc^{3+} leaching from Sc-TBAPy after GP photodegradation. This highlights the stability of Sc-TBAPy after long-term operations in an aqueous solution in the presence of ROS.

3. Also evaluated the metal leaching after the reaction, and these metal ions may also have impact on the degradation process.

Response. We thank reviewer 2 for the comment. ICP-OES analysis was conducted on the solution following an 8 hr GP photodegradation reaction with Sc-TBAPy. The result revealed the presence of 1.6 ppm of Sc^{3+} in the solution, indicating only a 0.42% Sc^{3+} leaching from Sc-TBAPy after GP photodegradation. This highlights the stability of Sc-TBAPy after long-term operations in an aqueous solution in the presence of ROS. Please see our response to reviewer 2’s comment 2.

4. The ROS were not related with the degradation pathways. The MS analyses of the byproduct normally provide evidence for the corresponding ROS.

Response. We thank reviewer 2 for their insightful comment regarding the relevance of degradation pathways to the involved ROS. In our investigation, we explored the pathways through which GP degrades and identified the cleavage of its C—N bonds as the predominant mechanism. Our findings indicate the involvement of three main ROS, $\bullet\text{OH}$, O_2^- and $^1\text{O}_2$, in the degradation process, resulting in a mixture of products. To elucidate the specific ROS responsible for each pathway, experiments were conducted using D_2O and in anoxic conditions. The results revealed that all three ROS play a role in driving the observed degradation pathways. The rate of formation of individual ROS appeared to favor a higher yield of one product over the other. This hypothesis was confirmed using electron spin resonance (ESR), which demonstrated that the higher rate of formation of $^1\text{O}_2$ species in Sc-TBAPy promotes the favorable generation of glycine and formic acid. In contrast, Al-TBAPy exhibited the formation of glycine, formic acid, and AMPA, attributed to the limited presence of $^1\text{O}_2$ species. This observation allowed us to propose that a higher concentration of $^1\text{O}_2$ species, as observed with Sc-TBAPy, is likely to result in the formation of products such as glycine, while a dominance of $\bullet\text{OH}$, and O_2^- species would drive the reaction toward the formation of AMPA, a toxic intermediate, as observed with Al-TBAPy. In addition, we investigated the roles of ROS using different scavengers and have incorporated our observations into our revised manuscript: “The role of the active ROS present in the oxidation of GP with Sc-TBAPy was further probed through the utilization of ROS scavengers. These scavengers, namely isopropanol (IPA), p-benzoquinone (p-BQ),

and sodium azide (NaN_3), were employed to quench $\bullet\text{OH}$ radical, $\text{O}_2^{\bullet-}$ radical and $^1\text{O}_2$ respectively. Observations derived from the experimental results indicated the complete disappearance of GP from solution, accompanied by the formation of glycine and formic acid. The relative concentrations were determined as 1:2 for p-BQ, 1:1 for IPA, and 1:2 for NaN_3 based on the integration of the corresponding peaks in the ^1H NMR spectra (Figure S30). These findings strongly suggest the substantial contribution of all three ROS to the oxidation of GP.”

In response to the comment from reviewer 2, “The MS analyses of the byproduct normally provide evidence for the corresponding ROS” we concur that MS analysis could offer additional support to our NMR findings. We acknowledge this suggestion and plan to pursue more research in future research. Meanwhile, we performed experimental analyses of the GP solution before and after the reaction using high frequency NMR magnets to gain insight into the reaction mechanism. Additionally, ESR analysis and scavenger quenching experiments with organic probes, along with reactions under D_2O and anoxic conditions, were performed to provide further understanding of the kinetics for GP degradation, and the active ROS in the reaction system. We are therefore confident that the information obtained from these methods is of sufficient quality to support and highlight the key findings of our studies.

5. Gentle reminder the EPR peak intensity could NOT advise the corresponding ROS pollution. The author need to use organic probes and kinetic calculations to perform the analysis.

Response. We thank reviewer 2 for their comment and suggestions. In addition to ESR analysis, we explored the influence of ROS by employing various organic probes as scavengers to observe their impact on the degradation of GP. Additionally, we agree with reviewer 2 regarding the insights that kinetic calculations could offer into the catalytic activity of the M^{3+} -TBAPy MOFs. To this note, we plan to consider this analysis into our future studies.

6. The evolution of different ROS should be revealed to gain more mechanistic insights.

Response. We appreciate this comment from reviewer 2, acknowledging the importance of elucidating the evolution of various ROS to enhance mechanistic understanding. Thus, we conducted photooxidation reactions on GP utilizing scavengers, namely isopropanol (IPA), p-benzoquinone (p-BQ), and sodium azide (NaN_3) to quench $\bullet\text{OH}$ radical, $\text{O}_2^{\bullet-}$ radical and $^1\text{O}_2$, respectively. In addition, our ESR analysis conducted under both dark and UV light conditions has confirmed the generation of distinct ROS driving the oxidation reaction. We believe that understanding the activity of ROS in the solution is crucial for delineating the reaction mechanism. We have summarized these findings in our revised manuscript: “...To further validate these findings, electron spin resonance (ESR) spectroscopy was performed on Sc-TBAPy and Al-TBAPy to monitor ROS formation. Under dark conditions, no ESR signal was observed for Sc-TBAPy and Al-TBAPy, indicating the absence of ROS formation. Upon light irradiation, $\bullet\text{OH}$, $\text{O}_2^{\bullet-}$ and $^1\text{O}_2$ were detected. The pertinent signal intensities in the $\bullet\text{OH}$, $\text{O}_2^{\bullet-}$ and $^1\text{O}_2$ spectra increased with longer irradiation from 1 to 2 min (Figures S28 and S29). Analysis of the intensity vs. time plots for the $\bullet\text{OH}$, $\text{O}_2^{\bullet-}$ and $^1\text{O}_2$ signals (Figure 6) revealed that Sc-TBAPy exhibits higher intensities for all detected ROS than Al-TBAPy, indicating a higher rate of ROS formation in Sc-TBAPy. Specifically, the formation yield of $^1\text{O}_2$ was the highest in both MOFs, while $\text{O}_2^{\bullet-}$ and $\bullet\text{OH}$ were observed at lower levels in Sc-TBAPy and Al-TBAPy. Based on ESR analysis, we concluded that the formation of $^1\text{O}_2$ favors the GP degradation, particularly with Sc-TBAPy, and likely contributes to the selective oxidation of GP. The non-selective catalysis observed with Al-TBAPy is likely due to the lower rate of $^1\text{O}_2$ formation and presence of $\text{O}_2^{\bullet-}$ and $\bullet\text{OH}$. The role of the active ROS present in the oxidation of GP with Sc-TBAPy was further probed through the

utilization of ROS scavengers. These scavengers, namely isopropanol (IPA), p-benzoquinone (p-BQ), and sodium azide (NaN₃), were employed to quench •OH radical, O₂⁻ radical and ¹O₂ respectively. Observations derived from the experimental results indicated the complete disappearance of GP from solution, accompanied by the formation of glycine and formic acid. The relative concentrations were determined as 1:2 for p-BQ, 1:1 for IPA, and 1:2 for NaN₃ based on the integration of the corresponding peaks in the ¹H NMR spectra (Figure S30). These findings strongly suggest the substantial contribution of all three ROS to the oxidation of GP.”

Reviewer 3:

In this paper entitled “Discovering a herbicide terminator: the role of metals in elimination glyphosate by Metal-Organic Frameworks” authors reported a 3 modified TBAPy MOF based on the Sc-TBAPy and studied their use in GP elimination from water (adsorption and photocatalysis). Although the important use of GP in our fields, it is an important contaminant of our environments. The authors demonstrated for the first time that MOFs can be used as catalyst in the degradation of GP. The text is clear and easy to read. However, I believe that the claims are high, and some more data and experiments are needed. Therefore, I recommend a major revision. It will be great to accept the manuscript after the demanded information, experiments, and changes. I strongly believe that this is a great work, but it can be superior.

Response. We appreciate the positive feedback from reviewer 3.

In my opinion, the introduction is quite extensive and impressive. But only a work is mentioned on the use of MOFs in GP elimination of water. Other articles should be also mentioned: <https://doi.org/10.1016/j.jpccs.2021.110403>, <https://doi.org/10.1021/acsomega.8b00921>, and <https://doi.org/10.1002/adfm.202213862>.

Response. We thank reviewer 3 for the comments and suggestions. We agree with reviewer 3’s suggestion that incorporating more examples of using MOFs for GP removal from water would improve our descriptions and motivation. That said, and to avoid having a long introduction (as per reviewer 1’s comment), we included the suggested works by reviewer 3: “*NU-1000, UiO-66, UiO-67, MIL-125, and MIL-101, have been shown as effective materials for GP adsorption.*^{58,67-69}” It is important to note that the examples mentioned primarily focus on the use of MOFs in GP adsorption rather than the photodegradation of GP. As there is limited research on the application of MOFs in GP photodegradation, we could only include one article to support this statement.

2. Probably authors should comment in the abstract that the degradation of GP is studied in water media.

Response. We thank reviewer 3 for their comment. We agree with reviewer 3’s suggestion and revised the abstract to state that our study investigates the degradation of GP in water. The abstract reads: “...*The impact of metals within a family of isostructural metal-organic frameworks (MOFs) on the adsorption and photodegradation of the herbicide glyphosate (GP) in water is presented in this study. Upon light irradiation for 5 min, Sc-TBAPy completely degrades 100% of GP in a 1.5 mM aqueous solution....*”

3. What is the initial concentration of GP in water in the elimination processes (adsorption and photodegradation)? As I see in the oxidation half reaction 1.5 mM of GP solution is used. Why authors have selected this concentration? Which is the GP concentration found in contaminated waters? The selected GP concentration in the elimination test should be in accordance with the normally concentration found in waters.

Response. We thank reviewer 3 for raising valid questions concerning the concentration of GP solution in our reactions. In our manuscript, we elaborate that the concentration of GP in the water employed for both the adsorption and photodegradation reactions was 1.5 mM. To justify this choice, we have incorporated the following text in our revised manuscript: “...*This concentration was determined suitable to analyze the concentration of GP as it decreases in solution.*” Lower concentrations of GP could not be adequately detected satisfactorily with NMR. The concentration utilized in our study exceeds typical GP concentration in water, where GP levels have been measured

up to 0.62 mM (105 ppm; *Environ. Sci. Pollut. Res.* **28**, 60635–60648 (2021)), with the acceptable limit set by the EPA at 0.7 ppm. The activity of our catalyst in degrading a 1.5 mM (845 ppm) GP solution indicates that its suitability for adsorbing GP at concentrations below 1.5 mM, and we plan to explore this in future studies.

3. Authors assess the stability of Sc-TBAPy after the photocatalytic process by different techniques: FTIR, DXRP and BET surface measurements. Interestingly the BET surface increased once the Sc-TBAPy was used in GP in UV-light and washed DMF. Probably some degradation of the framework occurs during this process (from 793 to 884 mg/g). Considering that the final application is water remediation, authors should check if there is some linker or metal on the final water solution after the GP degradation process.

Response. We thank reviewer 3 for their comments and valid suggestions. We agree that analyzing the water solution after the GP degradation process can offer additional insights into the stability of Sc-TBAPy following GP photodegradation. To explore this aspect, we conducted Inductively Coupled Plasma Optical Emission spectroscopy (ICP-OES) experiment to analyze the solution after an 8 hr photocatalytic GP degradation reaction with Sc-TBAPy. The results indicated the presence of 1.6 ppm of Sc³⁺ in the solution, corresponding to approximately 0.42% degradation of Sc-TBAPy. This suggests minimal leaching of Sc³⁺ from Sc-TBAPy during the degradation process. This slight Sc³⁺ leaching might potentially generate defects within the Sc-TBAPy structure, leading to an increase in the BET surface area of Sc-TBAPy. We have included these results, along with the corresponding discussion and instrumentation method for ICP-OES, in both the manuscript and SI.

4. Further, the recyclability of Sc-TBAPy in the photocatalytic tests should be done in order to assess this point. Is it Sc-TBAPy stable? How many cycles can we use Sc-TBAPy to degrade?

During the introduction authors mentioned some problems of the actually studied methods: 1) SELECTIVITY “there remains an essential need to develop selective photocatalysts”, 2) REUSABILITY “Post-treatment recovery of adsorbents remains challenging, which limits the ability to easily reuse these materials for GP capture”, and LAB SCALE 3) “most studies have been conducted at laboratory scales using controlled conditions, while high efficiency of GP removal required acidic conditions” In this work, there is no selectivity test, the reusability of the photocatalyst is not included and the test have been performed at lab scale. So, when compared the state of the art in the use of different materials in GP elimination from water and the final results I cannot see any reasons about the selection of MOFs in GP elimination. May be the introduction is highly ambitious and should be modified.

Response. We appreciate reviewer 3 for their comments. We reported the recyclability of Sc-TBAPy for degrading GP through three consecutive cycles, with no detectable GP remaining in solution. In our revised manuscript, we describe: “...Alternatively, when the reaction was conducted under light, no GP was detected in solution. The reaction under light was subsequently performed over three consecutive cycles, producing glycine and formic acid with no sign of GP in solution, as confirmed by ¹H NMR analysis (Figures S18b-S19). The absence of GP in solution indicates its complete degradation and the ability of Sc-TBAPy to photodegrade GP over three cycles.”

Regarding the selectivity of the Sc-TBAPy for GP degradation, we refer only to the photocatalyst's propensity to convert GP to specific products, specifically glycine, while avoiding the formation of AMPA. Our study highlights that the selectivity of the M³⁺-TBAPy MOFs is unique to Sc-TBAPy, as it forms products from the cleavage of the α C-N bonds of GP. In contrast, all other M³⁺-TBAPy

MOFs generated degradation products from both the cleavage of the α and β C-N bonds of GP. This preferential degradation observed in Sc-TBAPy suggests its unique selectivity in degrading GP.

Concerning the reusability of the MOF, our demonstration of Sc-TBAPy's reusability is based on the recyclability test conducted, as explained above. The stability of the MOF after recycling experiments was also confirmed through FTIR and BET surface area measurements.

In response to the reviewer's comment about most studies being conducted at the laboratory scale, we conducted experiments in natural river water and basic solution (pH 10.7). The results confirm that Sc-TBAPy can degrade GP in both natural river water or basic solutions. To confirm the stability of the recovered Sc-TBAPy under basic conditions, PXRD patterns demonstrated the recovered MOF's stability after irradiation in a pH 10.7 GP solution. The manuscript has been revised accordingly. *"...To investigate the impact of Sc-TBAPy in the degradation of GP in a natural environment, a 1.5 mM GP solution was prepared using sampled river water. Additionally, another GP solution with the same concentration was prepared, and its pH was adjusted to 10.7 using 1M NaOH to evaluate the activity of the MOF in an oxidizing environment. Subsequently, the reactions were performed. Results from ^1H NMR analysis revealed complete degradation of GP in the solution, leading to the formation of glycine and formic acid (Figure S23). To verify the stability Sc-TBAPy under basic conditions, its PXRD pattern was collected. The results demonstrated that the recovered Sc-TBAPy remained stable after exposure to irradiation in a pH 10.7 GP solution, indicating the superior activity of the MOF in both acidic and basic environments (Figure S24)."*

Additionally, we have removed the statement from the introduction that mentions the lab scale as a limitation since it does not relate to our specific contributions to the field.

REVIEWERS' COMMENTS

Reviewer #1 (Remarks to the Author):

The authors have made efforts to address the comments from the reviewers. I am generally satisfied with the changes they have made. one minor thing is a typo in Table 1. the radicals in the Al-TBAPy system in oxic conditions were not written correctly.

Reviewer #2 (Remarks to the Author):

The revised manuscript can be accepted in the present form

Reviewer #3 (Remarks to the Author):

Authors have successfully addressed all comments and changes. I strongly believe that the paper should be accepted for publication.